# On Pseudorandomness and Deep Learning: A Case Study

**Zahra Ebadi Ansaroudi** [1,*] ⓘ**, Rocco Zaccagnino** [2] ⓘ **and Paolo D'Arco** [2] ⓘ

1.   Center for Cybersecurity, Fondazione Bruno Kessler, 38122 Trento, Italy
2.   Dipartimento di Informatica, Universitá di Salerno, 84084 Fisciano, Italy
*   Correspondence: zebadiansaroudi@fbk.eu

**Abstract:** Pseudorandomness is a crucial property that the designers of cryptographic primitives aim to achieve. It is also a key requirement in the calls for proposals of new primitives, as in the case of block ciphers. Therefore, the assessment of the property is an important issue to deal with. Currently, an interesting research line is the understanding of how powerful machine learning methods are in distinguishing pseudorandom objects from truly random objects. Moving along such a research line, in this paper a deep learning-based pseudorandom distinguisher is developed and trained for two well-known lightweight ciphers, Speck and Simon. Specifically, the distinguisher exploits a convolutional Siamese network for distinguishing the outputs of these ciphers from random sequences. Experiments with different instances of Speck and Simon show that the proposed distinguisher highly able to distinguish between the two types of sequences, with an average accuracy of 99.5% for Speck and 99.6% for Simon. Hence, the proposed method could significantly impact the security of these cryptographic primitives and of the applications in which they are used.

**Keywords:** pseudorandomness; cryptographic primitives; deep learning; cryptanalysis

## 1. Introduction

Cryptographic primitives greatly rely on *true randomness*, a scarce and costly resource, and are evaluated often according to the quality of the *pseudorandomness* they produce. Therefore, in many cases, the challenge in realizing secure cryptographic primitives becomes finding design strategies which yield pseudorandom objects or behaviors. Indeed, many systems may fall apart if the underlying primitives miss this goal.

So far, several standards and test requirements for the pseudorandomness of cryptographic primitives have been presented, as well as related tools [1–4]. Roughly speaking, statistical tests should not be able to distinguish between a truly random source and the output of a cryptographic primitive.

Recently, deep learning (DL, for short) has made significant advances, leading some researchers to believe in its ability to reveal patterns in random-looking objects that were previously undetectable using conventional methods. In [5], the first usage of deep neural networks for testing the randomness of the outputs of the Speck lightweight block cipher was proposed. Therein, the pseudorandom distinguisher, obtained by combining neural networks with traditional cryptanalysis techniques, provided interesting results when compared to traditional techniques. Other works have been proposed in this direction, in which increasingly complex networks are trained on large datasets of sequences. However, the idea of facing the problem of distinguishing the outputs of ciphers from truly random sequences as a "binary classification problem" can suffer from some limitations, including the issue of how to manage the huge number of data available for the training and, consequently, the search for ever deeper and more complex supervised models to learn them. An alternative approach we pursue in this work could be to see the same problem as a problem of computing a "similarity function": if we are able to construct an efficient method to establish whether the output of a cipher is much more similar to another output

of the same cipher than to a truly random sequence (and vice versa), we will be able to de facto distinguish a similarity function.

In the DL area, Siamese networks have shown to be a powerful tool for tackling problems of similarity, with several applications since their introduction [6–8]. To date, no application to cryptanalysis, and specifically to the randomness evaluation of cryptographic primitives, has been proposed except for the one provided in [9], to defeat the privacy features of the Gossamer protocol. In this work, we propose a DL-based distinguisher which exploits a Siamese network on cryptographic primitives such as the Speck and Simon block ciphers. As we will see in Section 5, preliminary experiments with different instances of Speck and Simon on two datasets of 5000 sequences, one of 2500 random 32-bit sequences and the other one of 2500 cipher outputs, have been conducted. The results obtained show that the proposed distinguisher is able to achieve a high ability in distinguishing between the two types of sequences, with an average accuracy of 99.5% for Speck and 99.6% for Simon, improving on Gohr's earlier work [5]. For a description of Speck and Simon block ciphers, the interested reader is referred to Appendix A.

## 2. Pseudorandomness

In this section we introduce the fundamental notion of pseudorandomness. For further details and a more formal treatment, the reader can consult [10].

*Pseudorandom generator.* A pseudorandom generator is an efficient (deterministic) algorithm that, given short seeds, stretches them into longer output sequences, which are computationally indistinguishable from uniform ones. The term "computationally indistinguishable" means that no efficient algorithm, the distinguisher, can tell them apart [11–13].

*Pseudorandom function.* A pseudorandom function $F : \{0,1\}^n \times \{0,1\}^\ell \to \{0,1\}^n$ is an efficiently computable two-input function such that, for uniform choices of $k \in \{0,1\}^\ell$, the univariate function $F_k : \{0,1\}^n \to \{0,1\}^n$ is computationally indistinguishable from a univariate function $f : \{0,1\}^n \to \{0,1\}^n$, chosen uniformly at random from the set of all univariate functions of $n$-bit inputs to $n$-bit outputs.

A pseudorandom *permutation* can be defined similarly.

*Pseudorandom permutation.* A pseudorandom permutation $P : \{0,1\}^n \times \{0,1\}^\ell \to \{0,1\}^n$ is a two-input permutation such that, for uniform choices of $k \in \{0,1\}^\ell$, the univariate permutation $P_k : \{0,1\}^n \to \{0,1\}^n$ and its inverse $P_k^{-1} : \{0,1\}^n \to \{0,1\}^n$ are efficiently computable, and $P_k$ is computationally indistinguishable from a univariate permutation $p : \{0,1\}^n \to \{0,1\}^n$, chosen uniformly at random from the set of permutations on $n$-bit strings.

Often, in practice, block ciphers are evaluated depending on how well they approximate the behavior of a pseudorandom function or permutation.

*Distinguishing experiment.* Let $E : \{0,1\}^n \times \{0,1\}^{2n} \to \{0,1\}^n$ be a block cipher (in our case, it is Speck32 or Simon32 with block length $n = 32$ and key-size $\ell = 64$). Let $c = E(p,k)$ be a $n$-bit ciphertext. The description of the experiment in [9], Pseudo-R, defined for any distinguisher $D$ for the block cipher $E$, can be re-phrased as follows:

---

**Pseudo-R$_{D,E}$**

1. A bit $b$ is chosen uniformly at random.
   If $b = 1$, then a key $k \in \{0,1\}^{64}$ is chosen uniformly at random, and an oracle $\mathcal{O}(\cdot)$ is set to reply to queries using $E(\cdot, k)$. Otherwise, a function $e : \{0,1\}^{32} \to \{0,1\}^{32}$ is chosen uniformly at random, from the set of all the functions of 32 bits to 32 bits, and $\mathcal{O}(\cdot)$ is set to reply to queries using $e(\cdot)$.
2. $D$ receives access to oracle $\mathcal{O}(\cdot)$, and obtains replies to at most $t$ queries.
3. $D$ outputs a bit $b'$.
4. The output of the experiment is 1 if $b' = b$; otherwise, it is 0.

---

The block cipher $E$ is said to be $(t, \epsilon)$-pseudorandom if, for *any* distinguisher $D$, which runs for at most $t$ steps, there exists a small $\epsilon$, such that

$$Pr[\text{Pseudo-R}_{D,E} = 1] \leq 1/2 + \epsilon. \tag{1}$$

Roughly speaking, the block cipher is pseudorandom if any efficient distinguisher $D$ has no better distinguishing strategy than guessing (uniformly at random).

Several methods can be employed in order to set up an efficient distinguisher. For example, the distinguisher may employ differential cryptanalysis: given a plaintext pair $(P_0, P_1)$, with difference $\Delta P = P_0 \oplus P_1$, if the ciphertext difference $\Delta C = C_0 \oplus C_1$ is such that $Pr(\Delta P \xrightarrow{E} \Delta C) > 2^{-n}$, then $E$ is not a pseudorandom generator. Or, it may use other algorithmic techniques to find patterns, similarities or whatever can be useful for distinguishing. We are interested in this work in ML methods. Actually, ML methods are quite general and provide flexibility in implementing cryptanalytic strategies. Indeed, ML methods can also implement differential cryptanalysis, by tackling the techniques as a *binary classification* issue, where the ciphertext difference $\Delta C$ is a feature used in the training phase to define whether or not a given sequence is pseudorandom or not. Notice that our deep Siamese network correctly differentiates cipher outputs from random sequences without using a feature for ciphertext difference.

### 3. Previous Works

In 1991, Ronald Rivest presented a survey on the relationship between cryptography and the ML field, suggesting some directions for future cross-fertilization of the fields [14]. Since then, several attempts to use neural networks as a tool for designing cryptographic primitives and protocols have been proposed (e.g., [15,16]) as well as cryptography and, more precisely, multiparty computation and homomorphic encryption, have been used to set up secure and private implementations of neural networks, e.g., [17–20].

So far, few applications to cryptanalysis are also known, e.g., [5,21–23]. In this context, DL has made a noticeable progress in recent years for various difficult tasks, leading some researchers to believe that it can be used for detecting patterns in random-looking objects that were previously undetectable using conventional methods. Gohr's research [5], showed that DL can generate incredibly efficient cryptographic distinguishers: precisely, in his work, first, for 5 up to 8 rounds, a differential distribution table of round-reduced Speck32, with a given input difference, under the Markov assumption [24], was constructed; then, using residual neural networks and exploiting differential properties of Speck32, a chosen-plaintext attack on 9-round Speck32 was presented. As a result, in terms of classification accuracy, neural distinguishers outperformed multiple differential distinguishers that made use of the full distribution table. Hence, the work showed that the neural distinguishers use features that are invisible to any purely differential distinguisher, even when given unlimited data.

With regards to the cryptanalysis of Speck and Simon, other works followed. Zahednejhad et al. [25] applied the DL methodology proposed by Gohr to construct a neural-based integral distinguisher scheme for several block ciphers, including Speck32, Present, RECTANGLE, and LBlock. The neural-based integral distinguisher increased the number of distinguished rounds of most block ciphers by at least one round, when compared to the state-of-the-art integral distinguishing method. This clearly showed the potential of combining integral cryptanalysis and DL. Inspired by Gohr's work, Baksi et al [26] attempted to simulate differential cryptanalysis on non-Markov ciphers such as 8-round Gimli-Hash and 3-round Ascon-Permutation, showing that an attacker can use a multilayer perceptron (MLP, for short) and reduce the complexity of finding a pattern in the cipher outputs. In [27], the authors proposed a new technique and improved the performance of Gohr's neural distinguisher. Specifically, they employed an MLP rained on ciphertext differences rather than ciphertext pairs across Speck and Simon. In the case of Speck32, they achieved 98% of accuracy for distinguishing 9-round Speck. Given that data complexity is a major barrier to

practical key recovery attacks utilizing differential cryptanalysis, this novel technique is quite suitable in terms of round number and data complexity.

Hou et al., in [28], trained 8-round and 9-round differential distinguishers for Simon32, based on deep residual neural networks, and explored the impact of the input difference patterns on the accuracy of the distinguisher, with a success rate of over 90%. However, compared to the traditional approach, the time and data complexity of their DL-method are lower. Later on, Hou et al. [29] proposed a new method to improve neural distinguishers in terms of accuracy and number of rounds. Based on their new model, which is obtained with the help of the SAT/SMT solver [30], neural distinguishers for Simon and Speck were constructed. They enabled to better distinguish reduced-round Simon or Speck from a pseudorandom permutation.

However, all previous attacks leave it unclear which features or internal structures affect the success probabilities of those attacks. Benamira et al. [31] and Chen et al. [32] addressed this question by analyzing Gohr's findings, but they were unable to identify any DL-specific features that would affect the success probabilities of DL-based cryptanalysis, apart from the features related to linear dependencies or differential ones. Indeed, in [31], the authors showed that the neural distinguisher generally relies on the differential distribution of the ciphertext pairs, especially in the third and second-last rounds. The impact of features related to the cipher's round function was confirmed by [32]. Later on, Kimura et al. [33] were able to identify DL-specific features. They proposed DL-based output prediction attacks on several tiny block ciphers, including substitution-permutation based ciphers and Feistel-network based ones. Their work showed that, unlike linear/differential attacks, swapping or altering the internal parts of the target block ciphers has an impact on the average success probabilities of the proposed attacks.

Notice that the works briefly described have in common the idea to face the problem of distinguishing truly random sequences from outputs of ciphers as a "binary classification problem".

## 4. DL-Based Distinguisher

This section describes a distinguisher based on the one introduced in [9] but adapted to a 32-input feature vector, trained on two datasets of both random sequences and Speck32 (resp. Simon32) outputs. We start with a description of the Siamese network and of its learning method. Then, we present the DL-based distinguisher. For further details on Siamese networks, the interested reader is referred to [8].

### 4.1. Siamese Network

A Siamese network with two hidden layers and a logistic prediction $p$ is shown in Figure 1. As can be seen, the Siamese network consists of two identical sub-networks that run side by side, where each one takes an $N$-bit binary sequence $b_{i,1}, \ldots, b_{i,N}$ as input, for $i = 1, 2$. The twin networks share the weights at each layer (i.e., $w_{1,1}, \ldots, w_{N,M}$). Each network computes the features of its input, i.e., $(h_{1,1}, \ldots, h_{1,M})$ and $(h_{2,1}, \ldots, h_{2,M})$, respectively. Then, in the next layer, it uses a distance function, such as the $L1$ distance or the $L2$ distance, and calculates the distance $(d_1, \ldots, d_N)$ between the twin feature vectors. The similarity of these feature vectors is then computed based on the sum of their weighted distances, passed over an activation function. The output of the entire Siamese network, $p$, is, therefore, a *similarity measure* of the two inputs.

*One-shot learning.* Siamese networks are based on the so-called one-shot learning [8]. Such a learning strategy has the advantage that the network is able to classify objects even given only one training example for each category of objects. Indeed, during the training phase, instead of receiving pairs (object, class), the Siamese network receives (object1, object2, same-class/different-class ) as input and learns to produce a similarity score, denoting the chances that the two input objects belong to the same category. Typically, the similarity score is a value between 0 and 1, where the score 0 denotes no similarity, while the score 1 denotes full similarity.

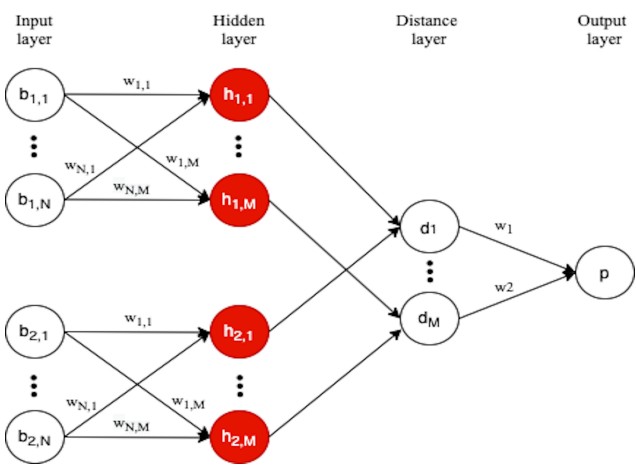

**Figure 1.** A simple Siamese neural network architecture.

*4.2. Model Definition*

Our proposal for a DL-based distinguisher is based on a convolutional Siamese neural network, whose architecture is depicted in Figure 2. It consists of two main parts: a *feature learning* part and a *similarity learning* part. For the sake of conciseness, in the description, a certain familiarity with neural networks is assumed. The reader with no background can consult any introductory textbook to the field.

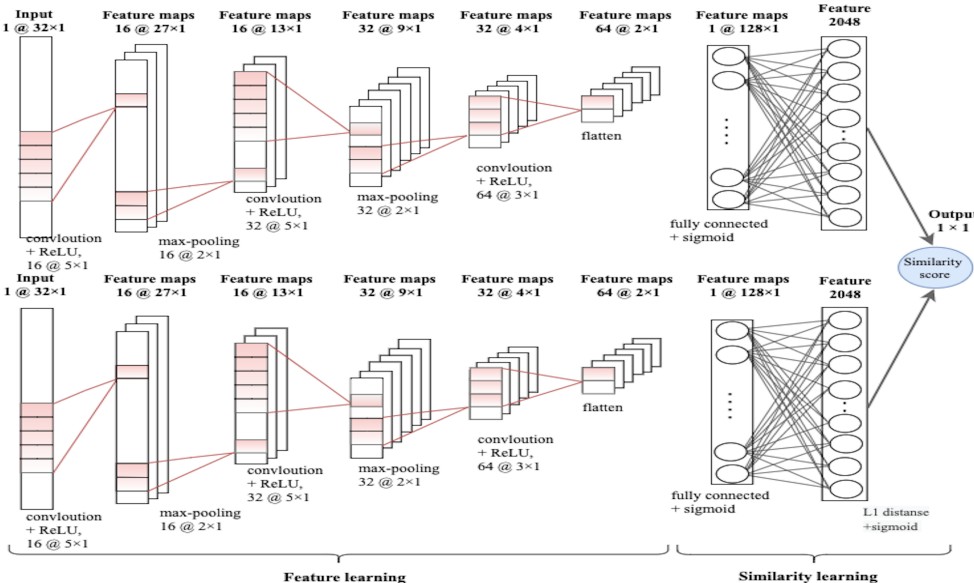

**Figure 2.** Our convolutional Siamese neural network architecture.

The feature learning part learns a function $c$, which maps our binary inputs into a high-level feature space. Let $s_i$ and $s_i'$ be the input sequences of the network, and let $F_i$ and $F_i'$ be the outputs of the feature learning part, that is, $F_i = c(s_i)$ and $F_i' = c(s_i')$. The function $c(\cdot)$ is implemented by a convolutional neural network (CNN, for short) [34,35]. Previous experience in the field, intuition, and trials and errors, guided us to the following choices: in each sub-network, our CNN architecture consists of three one-dimensional convolutional layers, with filters and kernels of varying sizes: in order, 16 filters with size $(5 \times 1)$, 32 filters with size $(5 \times 1)$, and 64 filters with size $(3 \times 1)$. The network applies a ReLU activation function to the output feature maps, followed by a one-dimensional max-pooling layer with size $(2 \times 1)$. Each layer of the sub-networks receives the output from its immediate previous layer as its input, and passes its output (as input) to the next layer. Higher-layer features are derived from features propagated from lower layers. As

the features propagate to the highest layer, the dimensions are reduced, depending on the size of the kernel. The number of feature maps is increased to better represent the input sequence and ensure similarity accuracy. The output of the final convolutional layer is then flattened into a single vector.

The similarity learning part learns how similar two input sequences are. Precisely, the output vectors of the feature learning part, $F_i$ and $F_i'$, are used as the inputs to the fully-connected feed-forward layers. The fully-connected layers are then followed by one more distance layer, which computes the L1 distance metric, by summing the absolute differences of the components of the vectors. The distance value is then passed to a single sigmoidal output unit, which maps the value into the interval $[0, 1]$. Score 0 denotes no similarity, while score 1 denotes full similarity.

*4.3. Loss Functions*

Loss functions are used, during the training process, for evaluating the model instances and selecting the most suitable one. Precisely, the loss functions quantify how well the siamese network made the correct decision. The lower the loss value, the better decisions were. Several loss functions have been proposed in the literature. In this work, both the *binary cross-entropy* loss function and the *contrastive* loss function have been exploited to train the network showed in Figure 2. The results obtained are provided in the next section.

A cross-entropy loss function is usually used in classification problems. It measures the difference between the desired probability distribution and the predicted probability distribution. In our specific context, *binary cross-entropy* is a valid choice because what we are essentially doing is *2-class classification*: *(i)* either the two sequences given as input to the network are of the same type, random or cipher output (i.e., class 1), *(ii)* or the two sequences are of different types (i.e, class 2). Binary cross-entropy loss is mathematically defined as:

$$binary\ cross\text{-}entropy\ loss = -\frac{1}{N} \sum_{i=1}^{N} y_i \log p_i + (1 - y_i) \log(1 - p_i), \tag{2}$$

where $y_i$ is the *label* of the $i$-th training sample (pair of sequences), i.e., $y_i = 1$ if the $i$-th training sample, given as input to the network, belongs to class 1; otherwise, $y_i = 0$. Furthermore, $p_i$ is the probability that the $i$-th training sample belongs to class 1, while $N$ denotes the number of training samples.

However, state-of-the-art Siamese networks exploit contrastive loss functions when training. Usually, these functions are better suited for such a kind of networks, and tend to improve accuracy. The idea is that the goal of a Siamese network is not to classify a set of sequence pairs but, instead, to differentiate them, by computing a similarity measure. Essentially, a contrastive loss function evaluates how good the Siamese network is in distinguishing between the sequence pairs. Contrastive loss is mathematically defined as:

$$contrastive\ loss = \frac{1}{N} \sum_{i=1}^{N} (1 - y_i) \times \frac{1}{2} d_i^2 + y_i \times \frac{1}{2} \{max(0, m - d_i)\}^2, \tag{3}$$

where $y_i$ is the *label* of the $i$-th training sample, that is, as in the previous case, $y_i = 1$ if the $i$-th training sample belongs to class 1; otherwise, $y_i = 0$. The value $d_i$ represents the Euclidean distance between the outputs of the two twin sub-networks, when given in input the $i$-th training sample, for $i = 1 \dots N$. Furthermore, $m > 0$ is a value called the *margin*. When $y_i$ equals 0, the amount of loss caused by similar pairs is quantified only by the first term and is minimized. Conversely, if $y_i$ equals 1, the loss is quantified only by the second term and is maximized by $m$. Thus, there is no loss when input pairs are non-similar and their distance exceeds $m$.

## 5. Experiments

To assess the effectiveness of our DL-method for distinguishing the outputs of *different instances* of Speck and Simon from random sequences, two ground datasets of 5000 samples have been considered for our experiments. Each ground dataset includes the same proportion of cipher outputs (Speck or Simon, respectively) and random sequences, i.e., a set of 2500 cipher outputs, computed by choosing random inputs and key values, and a set of 2500 random 32-bit sequences. Python's os.urandom() function has been used to generate the random numbers. The datasets are available on Github [36].

The following subsection describes the use of each ground dataset to create a dataset of similar and non-similar objects for training the Siamese network.

### 5.1. Datasets

Let $\mathcal{D}$ be a 5000-element ground dataset. We split $\mathcal{D}$ in the following disjoint subsets:

1.  $\mathcal{G}_{tr} \subseteq \mathcal{D}$, consisting of the 50% of samples randomly chosen (i.e., 2500 samples), and used as *training set generator*, i.e., to build the *training set* for the Siamese network.
2.  $\mathcal{G}_{vd} \subseteq \mathcal{D}$, consisting of the 30% of samples randomly chosen (i.e., 1500 samples), and used as *validation set generator*, i.e., to build the *validation set* for the Siamese network.
3.  $\mathcal{G}_{te} \subseteq \mathcal{D}$, consisting of the remaining 20% of samples (i.e., 1000 samples), and used as *test set generator*, i.e., to build the *test set* for the Siamese network.

The *training set* $\mathcal{T}$ is formally defined as $\mathcal{T} = \mathcal{T}_1 \cup \mathcal{T}_0$, where:

$$\mathcal{T}_1 = \{(s_i, s_j) \mid s_i, s_j \in \mathcal{G}_{tr}, l(s_i, s_j) = 1\}, \mathcal{T}_0 = \{(s_i, s_j) \mid s_i, s_j \in \mathcal{G}_{tr}, l(s_i, s_j) = 0\}$$

where $l(s_i, s_j) = 1$ (resp. $l(s_i, s_j) = 0$) indicates that $s_i$ and $s_j$ are similar (resp. non-similar). We remark that $\mathcal{T}_1$ (resp. $\mathcal{T}_0$) is a set of pairs of similar (non-similar) sequences from $\mathcal{G}_{tr}$.

In our experiments, we have generated $\mathcal{T}$ such that $|\mathcal{T}| = 10,000$ in two different ways:

*   200-$\mathcal{G}_{tr}$: $\mathcal{T}_1 = \mathcal{R}_1 \times \mathcal{R}_1 \cup \mathcal{C}_1 \times \mathcal{C}_1$, and $\mathcal{T}_0 = \mathcal{C}_2 \times \mathcal{R}_2 + \mathcal{R}_2 \times \mathcal{C}_2$, where $\mathcal{R}_i \subseteq \mathcal{G}_{tr}$ (resp. $\mathcal{C}_i \subseteq \mathcal{G}_{tr}$) consists of *random* (resp. *cipher output*) sequences randomly chosen from $\mathcal{G}_{tr}$, with $|\mathcal{R}_i| = 50$ (resp. $|\mathcal{C}_i| = 50$), for $i = 1, 2$. Observe that $|\mathcal{R}_1| + |\mathcal{R}_2| + |\mathcal{C}_1| + |\mathcal{C}_2| = 200$.
*   300-$\mathcal{G}_{tr}$: $\mathcal{T}_1 = \mathcal{R}_1 \times \mathcal{R}_1 \cup \mathcal{C}_1 \times \mathcal{C}_1$, and $\mathcal{T}_0 = \mathcal{C}_2 \times \mathcal{R}_2 + \mathcal{R}_3 \times \mathcal{C}_3$, where $\mathcal{R}_i \subseteq \mathcal{G}_{tr}$ (resp. $\mathcal{C}_i \subseteq \mathcal{G}_{tr}$) consists of *random* (resp. *cipher output*) sequences randomly chosen, with $|\mathcal{R}_i| = 50$ (resp. $|\mathcal{C}_i| = 50$), for $i = 1, 2, 3$. Observe that $|\mathcal{R}_1| + |\mathcal{R}_2| + |\mathcal{R}_3| + |\mathcal{C}_1| + |\mathcal{C}_2| + |\mathcal{C}_3| = 300$.

### 5.2. Experimental Setting

After creating one dataset for each cipher, the network has been trained for 1000 epochs, with a batch size of 24, and using the *Adam* optimizer [37], with a learning rate of 0.00005, and with the two chosen loss functions: the *binary cross-entropy* loss function and the *contrastive* loss function. Then, the network was validated using 1000 different one-shot learning tasks. Specifically, at every 10 epochs (one-shot evaluation interval), for every one-shot task, our model first chooses a *test sequence* $s_1$ from the validation set. Then, it creates two pairs, a similar pair and a non-similar pair, as follows:

*   the similar pair $< s_1, s_2 >$, includes the test sequence $s_1$ and a sequence $s_2$ from the same category,
*   the non-similar pair $< s_1, s_2' >$, includes the test sequence $s_1$ and a sequence $s_2'$ from the other category.

The obtained two pairs are used to compare the same object with two different ones, out of which only one of them is in the same category. For each of these two pairs, the network generates a similarity score, denoted with $S_1$ and $S_2$. Now, if the model is trained properly, then $S_1$ is expected to be the maximum. If it happens, then it is treated as a correct prediction; otherwise, it is an incorrect one.

Repeating this procedure for $\ell$ different one-shot tasks, the accuracy is computed as

$$\text{one-shot accuracy} = ncorrect/\ell, \tag{4}$$

where $\ell$ equals 1000, and *ncorrect* is the number of correct predictions out of the $\ell$ tasks.

*5.3. Results*

As mentioned, the validation set is evaluated every 10 epochs, with the best model being chosen to be tested on the test set at the end. This experiment was carried out a total of 10 times. The validation and train accuracies for the last run of the experiment using a binary cross-entropy loss defined by Equation (2) on the Speck dataset are shown in Figure 3.

As shown in Figure 3a, for this run of the experiment with a dataset built using 200 different samples, the validation accuracies range from 50% to 85%, and converge to 85% within the first 600 epochs, therefore choosing the model with higher validation accuracy as the final model for evaluation on the test set, results in a test accuracy of 83%. However, as shown in Figure 3b, with 300 different samples, the Siamese network outperforms the one using 200 samples. Hence, 300 different samples are our choice for building the datasets. The accuracies of 10 runs of the experiment, for 1000 different one-shot tasks on the test set using the binary cross-entropy loss, are computed according to Equation (4) and are provided in Table 1. As indicated, they resulted in an approximate average accuracy of 94% for both the Speck and Simon ciphers.

Based on the above findings, going back to our main goal, we can set up the following distinguisher $D$ for the experiment Pseudo-R$_{D,E}$:

---

**Distinguisher $D$**

1.  Builds the dataset as described before and selects the best DL-Model to distinguish (Simon/Speck) from a random permutation.
2.  Sends a query $x$ to the oracle $\mathcal{O}(\cdot)$, obtaining in response $y = \mathcal{O}(x)$.
3.  Chooses $y'$ uniformly at random.
4.  Constructs the pair $(y, y')$ and gives it in input to the DL-Model.
5.  If the DL-Model finds $y$ and $y'$ similar, then $D$ gives in output 0 (i.e., $\mathcal{O}(\cdot)$ implements a random permutation); otherwise, $D$ gives in output 1 (i.e., $\mathcal{O}(\cdot)$ implements Simon or Speck).

---

The distinguisher $D$, a one-query distinguisher, succeeds with probability:

$$\begin{aligned} Pr[\text{Pseudo-R}_{D,E} = 1] &= Pr(b=0) \cdot Pr(\mathcal{D} \text{ outputs } 0|b=0) + \\ &\quad + Pr(b=1) \cdot Pr(\mathcal{D} \text{ outputs } 1|b=1) \\ &= \frac{1}{2} \cdot 0.94 + \frac{1}{2} \cdot 0.94 > \frac{9}{10}. \end{aligned}$$

As a result, according to Equation (1), $D$ succeeds with an advantage of at least $\frac{4}{10}$ over a random guess. Hence, both ciphers do not yield a pseudorandom behavior. Note that a dataset of 5000 sequences is considered for the experiment, however, only 300 samples are chosen to create a training dataset of 10,000 elements for the Siamese network, for each experiment run. Considering 10 different runs of the experiment, at least $t = 3000 \approx 2^{11}$ steps are required to set up the best DL-Model. A similar conclusion applies to the cases of the contrastive loss function, presented in the following.

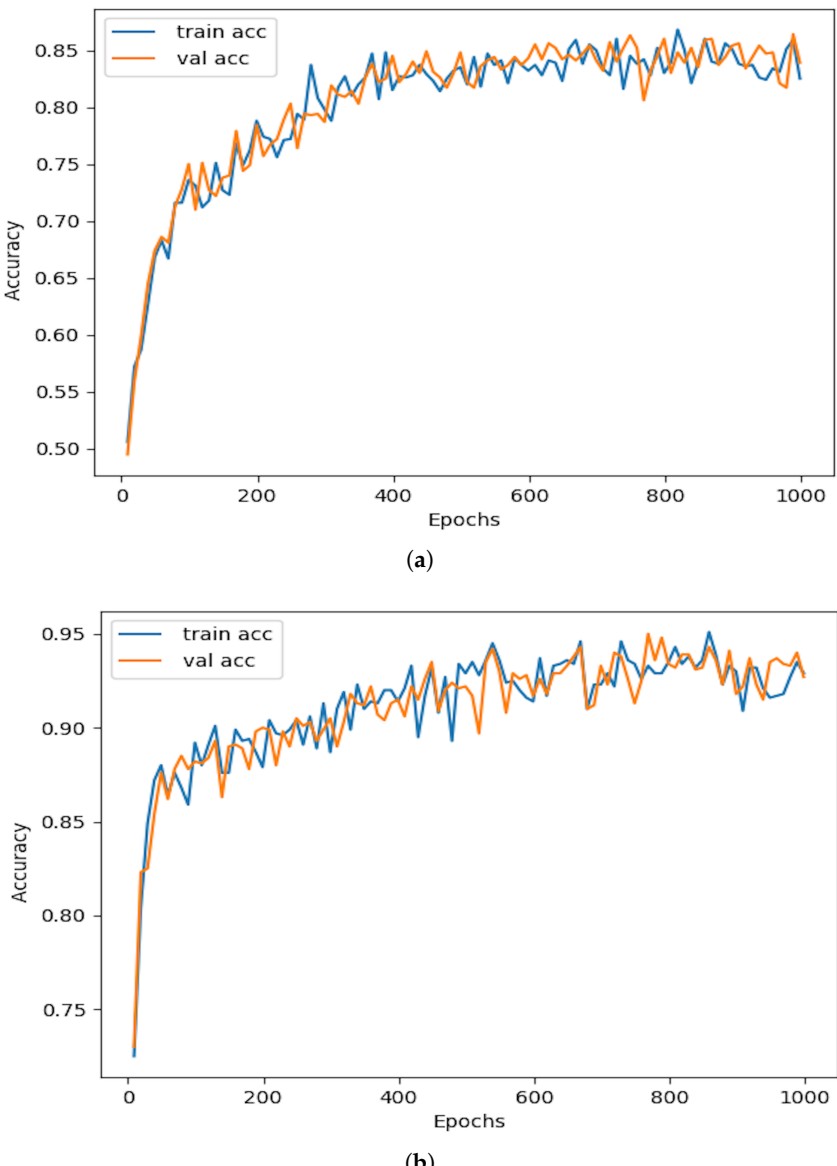

(a)

(b)

**Figure 3.** One-shot training and validation accuracy over epochs, for Speck, with a binary cross entropy loss. (**a**) With 200 samples, test accuracy 83%. (**b**) With 300 samples, test accuracy 91%.

**Table 1.** One-shot test accuracies with a binary cross-entropy loss, $Speck_n$ indicates "on Speck dataset with $100 * n$ numbers of samples".

| Cipher | 1 | 2 | 3 | 4 | 5 | 6 | 7 | 8 | 9 | 10 | Avg |
|--------|------|------|------|------|------|------|------|------|------|------|-------|
| $Speck_2$ | 0.81 | 0.76 | 0.82 | 0.9 | 0.87 | 0.84 | 0.8 | 0.74 | 0.87 | 0.83 | 0.82 |
| $Speck_3$ | 0.93 | 0.94 | 0.91 | 0.92 | 0.97 | 0.95 | 0.97 | 0.93 | 0.97 | 0.91 | 0.94 |
| Simon | 0.92 | 0.95 | 0.92 | 0.93 | 0.95 | 0.93 | 0.96 | 0.97 | 0.98 | 0.91 | 0.942 |

The experiment was repeated once more for a total of ten times, using a contrastive loss, as defined by Equation (3) with a margin value of 0.3. Table 2 provides the accuracy results. These findings show that a Siamese network with a contrastive loss outperforms the one with cross-entropy loss, and it reaches an average test accuracy of over 99% for both ciphers. Figure 4 shows the train and validation accuracies for the last run of the experiment on the Speck dataset. As it is clear for the first 100 epochs, accuracies are 100%

or close to it, and as the number of epochs increases, the accuracy drops to 84%. Thus, the best model is chosen at the first epochs.

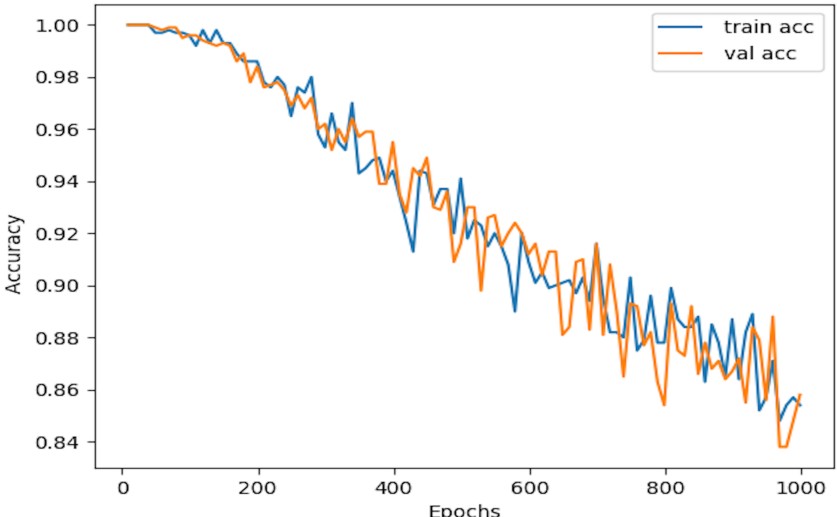

**Figure 4.** One-shot training and validation accuracy over epochs, for Speck, with a contrastive loss, given the evaluation accuracy of 100%.

**Table 2.** One-shot test accuracies with a contrastive loss.

| Cipher | 1 | 2 | 3 | 4 | 5 | 6 | 7 | 8 | 9 | 10 | Avg |
|--------|---|---|---|---|---|---|---|---|---|----|-----|
| Speck | 1 | 0.99 | 1 | 1 | 0.99 | 0.99 | 0.99 | 0.99 | 1 | 1 | 0.995 |
| Simon | 1 | 1 | 1 | 0.99 | 1 | 1 | 0.99 | 0.99 | 0.99 | 1 | 0.996 |

The obtained findings demonstrate that our proposed deep learning distinguisher, based on a Siamese network with a contrastive loss and the one-shot learning technique, provides an accurate solution for pseudorandomness evaluation. Our best models achieve an average accuracy of 99.5% for Speck, and 99.6% for Simon. In the case of Speck, Gohr's approach reached an accuracy of roughly 93% for five rounds of Speck.

As a follow-up research plan, the above findings encourage us to employ Siamese networks to differentiate among various ciphers. To begin, the distinguishability of Speck from Simon has been investigated and, using a binary cross-entropy loss, Speck and Simon outputs are distinguishable with an average accuracy of 94.4% (on 10 experiments, with accuracies in each experiment equal to 0.96, 0.93, 0.94, 0.94, 0.96, 0.95, 0.96, 0.95, 0.93, 0.92). Additional experiments, for example focusing on mixed ciphers, that connect the outputs of the Speck round function to Simon's inputs, or the other way around, could be helpful in assessing the effectiveness of Siamese networks. Future works might analyze such variants.

## 6. Conclusions

This paper investigated how to build a DL-based distinguisher for Simon32 and Speck32. Indeed, the two lightweight block ciphers are commonly used in several applications, e.g., in multi-round identification, authentication, and access control schemes, especially in computational limited environments, populated by cheap and resource-constrained devices. Assessing their security is therefore crucial. Although a convolutional Siamese network has proven effective for our task, it is critical to provide adequate training samples to the network (similar and non-similar pairs). For network optimization, two different loss functions have been considered: a binary cross-entropy loss and a contrastive loss function. With a data complexity of $2^{11} \approx 3000$, the network with a binary cross-entropy loss has shown an average test accuracy of 94%. The contrastive loss outperformed the binary cross-entropy loss, and gave a higher evaluation accuracy. The results of this paper seem to

be quite promising. They need some further evaluation on other primitives, on different datasets and, maybe, on different parameters. However, if the trend is confirmed, Siamese networks would be configured as an important tool to consider in future cryptanalytic tool packets.

**Author Contributions:** Methodology , Z.E.A. and R.Z.; Writing—original draft, Z.E.A.; Writing—review & editing, Z.E.A., R.Z. and P.D.; Supervision, P.D. All authors have read and agreed to the published version of the manuscript.

**Funding:** This work was partially supported by project SERICS (PE00000014) under the NRRP MUR program funded by the EU-NGEU.

**Institutional Review Board Statement:** Not applicable.

**Informed Consent Statement:** Not applicable.

**Data Availability Statement:** The datasets generated for this study are available at [36].

**Conflicts of Interest:** The authors declare no conflict of interest.

## Appendix A

**Simon and Speck.** In 2013, the National Security Agency (NSA) published the Speck and Simon block cipher families. These block ciphers are intended to provide security on limited devices with a focus on design simplicity [38].

There are 10 different versions for each cipher family based on the block and key size combinations (as shown in Table A1), making them suitable for a broad range of applications. $\text{Speck} 2n/mn$ will denote Speck with $n$-bit block size and $mn$-bit key size. A similar notation is used for Simon. Operations used by Simon and Speck are: modular addition and subtraction $\pm \mod 2^n$, bitwise xor $\oplus$, and &, left circular shift $<<<\ m$ and right circular shift $>>>\ m$ by $m$ positions. Modular addition and bitwise and & bring non-linearity and confusion for Speck and Simon, respectively, whereas diffusion is achieved using cyclic rotation and xor.

Any cipher version shares the same Feistel-structure and round function as shown in Figures A1 and A2.

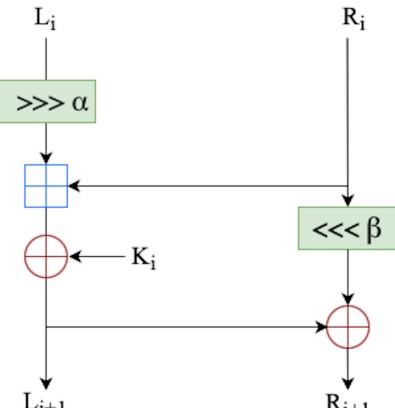

**Figure A1.** Speck round function.

The Feistel-structure runs the round function a fixed number of times, depending on the block and key size (presented in Table A1). Each round function takes as input two $n$-bit intermediate ciphertext words, $L_i$ and $R_i$, and an $n$-bit round key $K_i$. Then, it applies the round function and outputs two words $L_{i+1}$ and $R_{i+1}$, that are the input words for the next round. The outputs of the last round represent the ciphertext.

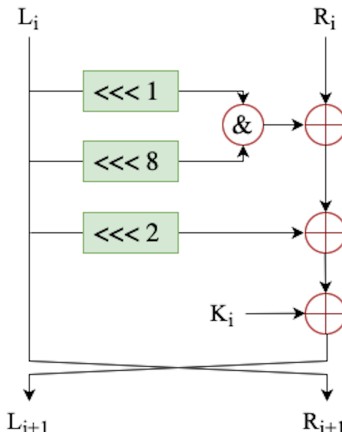

**Figure A2.** Simon round function.

**Table A1.** Variants of Speck and Simon. The parameters $\alpha$ and $\beta$, as seen in Figure A1, count the number of positions for the right and left shifts, respectively. The values $\alpha = 7$ and $\beta = 2$ are for Speck32/64, while other variants of Speck use $\alpha = 8$ and $\beta = 3$.

| Block Size | Key Size | Speck Rounds | Simon Rounds |
|---|---|---|---|
| 32 | 64 | 22 | 32 |
| 48 | 72, 96 | 22, 23 | 36, 36 |
| 64 | 96, 128 | 26, 27 | 42, 44 |
| 96 | 96, 144 | 28, 29 | 52, 54 |
| 128 | 128, 192, 256 | 32, 33, 34 | 68, 69, 72 |

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
