# Peer review of "On Pseudorandomness and Deep Learning: A Case Study"

_applsci, doi:10.3390/app13053372_

Round 1
Reviewer 1 Report
Ln. 74. ‘The definition also requires F −1 to be efficiently computable’ has been written. It should be elaborate to convince the readers. A reference is at least needed.
Ln. 82 Please see the below. This could be more attractive if authors could algorithmically describe the steps, as the algorithm below.
Author Response
Response 1: A reference added in Ln 56. Additionally, Lns. 62 to 73 provide further clarification on the concept.
Response 2: The picture is not opened.
Reviewer 2 Report
In this article, authors develop a deep-learning based pseudorandom distinguisher, which a Siamese network is trained to distinguish the outputs of two well-know lightweight ciphers, Speck and Simon. With high accuracy, such method may provide a significant criterion for the securities of these above-mentioned ciphers. Furthermore, this work may inspire adversarial networks to improve the computationally indistinguishable properties of pseudorandom generators. In my opinion, this work is interesting. I would like to recommend the publication of this manuscript. However, I have several concerns, so a revision is necessary before its publication. The following points should be clarified or revised.
(1) There are several steps involving random numbers, such as identifiers uniformly at random in the experiment Pseudo-R and random sequences in the ground datasets. I wonder whether such steps introduce extra influences of pseudorandomness to the results. Otherwise, the data sources of random numbers should be given.
(2) If the outputs of the Speck round function are connected to the inputs of Simon one, some mixed cipher outputs can be obtained. Is the trained Siamese network effective for similar complex situations?
(3) To improve the compactness of Fig. 6, signs of bitwise xor may be rearranged to one column, like the previous work [Fu et al, New integral attacks on SIMON, IET Inf. Secur. 11(5), 277-286 (2017)].
Author Response
Response 1: Ln. 247 contains reference datasets as well as the function used to generate random sequences.
Response 2: We didn't study mixed ciphers. It will be taken into account for future works as outlined in Lns. 334–336.
Response 3: Fig. 6 has been improved as suggested.
Reviewer 3 Report
In the introduction section, there is no need for a "roadmap of the paper"; just the content is enough as the last paragraph.
Define the Pseudorandom permutation also.
Provide equation numbers and refer to them in the text.
Give the reference for Adam optimizer.
Check for English language grammatical errors.
Author Response
Response 1: The roadmap paragraph has been removed.
Response 2: Added in Lns. 68-73.
Response 3: The equations have been numbered.
Response 4: The reference add in Ln. 272.
Response 5: The paper writing has been checked for this.
Reviewer 4 Report
1. Reference for One-shot learning, Siamese network and Adam optimizer etc may be included
2. How Distance layer values are decided in Figure 1?
3. Mention the Dataset reference in experiment section
4. Equations may be numbered
Author Response
Response 1: There was a reference for the siamese, that could also be used for a one-shot. We included that in Ln. 173.
Response 2: Lns. 166–172 now provide the description.
Response 3: Datasets reference has been added in Ln. 248, the experiment section.
Response 4: Equations have been given numbers.